# Correlation between the Surface Undulation and Luminescence Characteristics in Semi-Polar $\left(11\bar{2}2\right)$ InGaN/GaN Multi-Quantum Wells

**DOI:** 10.3390/nano13131946

**Published:** 2023-06-27

**Authors:** Mi-Hyang Sheen, Yong-Hee Lee, Jongjin Jang, Jongwoo Baek, Okhyun Nam, Cheol-Woong Yang, Young-Woon Kim

**Affiliations:** 1Research Institute of Advanced Materials, Department of Materials Science and Engineering, Seoul National University, Seoul 08826, Republic of Korea; mihyang.sheen@samsung.com (M.-H.S.);; 2Department of Nano-Optical Engineering, LED Technology Center, Tech University of Korea, Siheung-si 15073, Republic of Korea; 3School of Advanced Materials Science & Engineering, SungKyunKwan University, Suwon 16419, Republic of Korea

**Keywords:** semi-polar GaN, TEM-cathodoluminescence, indium concentration, surface undulation

## Abstract

Surface undulation was formed while growing InGaN/GaN multi-quantum wells on a semi-polar m-plane (1–100) sapphire substrate. Two distinct facets, parallel to $\left(11\bar{2}2\right)$ and $\left(01\bar{1}1\right)$, were formed in the embedded multi-quantum wells (MQWs). The structural and luminescence characteristics of the two facets were investigated using transmission electron microscopy equipped with cathodoluminescence. Those well-defined quantum wells, parallel and slanted to the growth plane, showed distinct differences in indium incorporation from both the X-ray yield and the contrast difference in annular darkfield images. Quantitative measurements of concentration in $\left(01\bar{1}1\right)$ MQWs show an approximately 4 at% higher indium incorporation compared to the corresponding $\left(11\bar{2}2\right)$ when the MQWs were formed under the same growth condition.

## 1. Introduction

Conventional InGaN/GaN multi-quantum wells (MQWs) grown on the c-axis of the corundum structure exhibit strong spontaneous and piezoelectric polarization fields [1]. These internal fields cause spatial separations of electrons and holes in the MQWs, hampering radiative recombination of the carriers, known as the quantum-confined stark effect (QCSE) [2]. Since this QCSE deteriorates the emission efficiency of light-emitting diodes (LEDs), many groups have attempted to fabricate non-polar [3,4] and semi-polar [5,6]. GaN epi structures to reduce the internal electric fields along the growth direction. Among various semi-polar planes, the $\left(11\bar{2}2\right)$ plane has recently attracted great interest because of its high indium incorporation capability and the superior crystal quality of InGaN/GaN MQWs with low internal polarization-induced electric fields [3,4,7,8,9].

However, semi-polar $\left(11\bar{2}2\right)$ GaN thin films contain a high density of extended defects [10,11,12] and exhibit anisotropic properties [13,14,15] due to the lack of crystal symmetries, delivering characteristic properties of optical gain anisotropy [13] and anisotropic strain [14]. In addition, the surface morphology of the epitaxial layer exhibits undulations accompanied by arrowhead-like features pointing toward $\left[11\bar{2}\bar{3}\right]$ [16,17,18]. The origin of this undulation remains unknown but is commonly attributed to the anisotropy originating from the difference in the mobility and diffusional length of adatoms on specific crystallographic planes, combined with the lattice mismatch of the semi-polar GaN film [15].

Undulated structures grown on semi-polar substrates exhibit some variation depending on growth conditions. The majority of the growth facets are $\left(10\bar{1}1\right)$ and $\left(10\bar{1}\bar{1}\right)$ planes when grown on the $\left(1\bar{1}02\right)$ plane of Al_2_O_3_ or bulk GaN [19,20]. Smooth undulation or mixed facets have also been reported in some thin films grown on semi-polar substrates [18,21].

Site-specific luminescence characteristics can provide critical data related to indium incorporation, which is a major concern for high efficiency and characteristic wavelength manipulations in LED. Significant efforts have been made to correlate the luminescence and microstructure in the undulated structure of InGaN/GaN MQW using photoluminescence [13,22], cathodoluminescence in scanning electron microscopy [10,23,24], electroluminescence, and near-field microscopy [25,26].

Transmission electron microscopy with cathodoluminescence (TEM-CL) can provide a superior spatial resolution of both images and resolved wavelengths to photoluminescence [13,14,15], with the capability to map the microstructure, chemical information, and luminescence wavelengths. This information can be used to directly correlate the quantum well thicknesses, defects, and indium incorporation probabilities. Herein, the anisotropic luminescence characteristics of InGaN/GaN MQWs grown on a semi-polar $\left(11\bar{2}2\right)$ substrate were investigated using cathodoluminescence combined with chemical distributions, crystallographic construction, and atomic-resolution microscopy.

## 2. Materials and Methods

The $\left(11\bar{2}2\right)$ semi-polar GaN epi was grown on a hemispherical patterned *m*-plane sapphire substrate (HPSS) by metal-organic chemical vapor deposition [27]. The LED structure was composed of an n-type GaN with a 2-μm-thick Si-doped layer and an active region consisting of five periodic quantum wells with 6-nm GaN barriers and 2.3-nm InGaN wells, followed by a 150-nm-thick p-type GaN:Mg layer. The doping concentration of the n-type GaN layer was about 5 × 10^18^ cm^−3^, and the hole concentration was estimated to be about 6 × 10^17^ cm^−3^. Details of the growth conditions, and the electro-luminescence characteristics can be found in the earlier report [27]. After the growth, thermal annealing was performed at 800 °C for 5 min under an N2 atmosphere.

For electron beam transparency, a cross-sectional TEM specimen was prepared using conventional mechanical thinning and ion milling (PIPS, Gatan, Pleasanton, CA, USA). The reference plane of the cross-sectional view was chosen so that the undulated surface ridge was parallel to the TEM viewing direction. From this viewing direction, the correlation between surface undulation and internal MQW structure was directly observed.

CL spectra and maps of the specimens were acquired using a home-built cathodoluminescence stage for TEM (JEM-2010F, JEOL, Akishima, Japan) with an analytical polepiece. The stage can collect X-ray signals for Energy Dispersive Spectroscopy (EDS) and cathodoluminescence simultaneously, with the capability of collecting signals over the 500 µm range. Spectra collection was performed at a low electron-beam acceleration voltage (120 kV) to reduce the electron beam damage and enhance the cross-sectional area in the scattering process. The specimen was cooled to 137 K to improve the radiative recombination rate and reduce the carrier diffusion length.

High-resolution (HR) and annular dark-field (ADF) STEM images were acquired using a Cs-corrected STEM (ARM-200F, JEOL, Japan) operated at 200 kV. The instrument was equipped with an aspherical aberration corrector (CEOS GmbH, Heidelberg, Germany) and an EDS detector (X-Max, Oxford Instruments, Oxford, UK).

## 3. Results

Figure 1a shows a cross-sectional annular dark field scanning transmission electron microscopy (ADF STEM) image of the $\left(11\bar{2}2\right)$ semi-polar LED structure grown on the hemispherical patterned sapphire substrate (HPSS) observed along [11-2-3]. Large voids were observed above the hemispherical-patterned sapphire substrate, which did not disturb the growth of single crystalline GaN. The surface undulation widths varied from position to position, as seen in Figure 1a, where there was no clear correlation between the substrate pattern and surface undulation lengths and periods. The undulation geometry of the sampling for the TEM analysis is schematically presented in Figure 1b along with the viewing direction. Surface undulation was widely observed and usually reported as arrow-like features [14,17,18,19] or striation [15].

Figure 1c shows a cross-sectional brightfield (BF) image revealing that the InGaN/GaN MQWs follow the contour of the surface undulations from a certain depth from the surface, which contains two distinct layers: flat—parallel to the substrate surface, and slanted—growth plane offset from the substrate surface, which is marked with a straight white dotted line. Figure 1d shows a panchromatic CL map of the data presented in Figure 1c from the 350 to 550 nm spectrum window. Even though the panchromatic emission of the MQWs seems uniform over the undulated structure, four distinct wavelengths were detected that originated from two different crystallographic planes.

Figure 1e shows the point CL spectra of the two crosshairs labeled GaN-A and GaN-B in Figure 1d. The peak wavelength obtained from GaN-A was approximately 365 nm, corresponding to an emission wavelength of the near band edge (NBE) and basal plane stacking faults (BSFs) [28,29]. NBE and BSF peaks are commonly observed in epitaxially grown $\left(11\bar{2}2\right)$ GaN. The signal from GaN-B showed a peak wavelength of approximately 374.5 nm, which is known to be related to the emission wavelength of partial dislocations (PDs) and prismatic stacking faults (PSFs) [28,29]. The donor-acceptor pair (DAP) luminescence may also coexist in this wavelength region.

Figure 1f shows the point CL spectra obtained from the flat and slanted MQWs. The peak emission wavelength of the flat MQWs was approximately 416 nm, while the luminescence of the slanted MQWs was composed of two maxima near 458 and 478 nm. The slanted MQWs exhibited longer wavelength emissions compared to those of the flat MQWs, and the peaks exhibited a broad full width at half-maximum.

Figure 2 shows the CL intensity maps for selected wavelength windows. Figure 2a shows the emission map of NBEs and BSFs with a wavelength window of 360–370 nm. In this specimen, BSF formed on oblique planes, angled 29° from the viewing direction, which is an angle between the $\left(0001\right)$ basal plane and $\left(11\bar{2}2\right)$ growth plane. Because the segmented BSFs overlap when viewed from the [11-2-3] direction, emission from BSFs overlapped throughout the foil thickness of the TEM sample, resulting in a non-uniform contrast. In addition, the carriers can likely diffuse within the QW plane generated by the SFs instead of being strongly confined, resulting in the broad distribution of the luminescence map. The upper part of the epitaxial structure, approximately 2 μm from the surface, exhibited brighter luminescence than the lower part because n-type doped GaN enhanced the DAP luminescence.

Figure 2b,c show CL intensity maps from MQWs with peak emissions at 410–425 and 445–485 nm, respectively. Figure 2d shows a red-green-yellow composite color map of the characteristic CL wavelengths. Red indicates the emission of slanted MQWs, green is the emission of flat MQWs, and yellow is the emission of NBE and BSFs. Figure 3e shows an enlarged image of the luminescence map overlapped with the ADF STEM image, clearly indicating that the flat MQWs emit a shorter wavelength than the slanted MQWs. These results are the first observations of different luminescence from an undulated GaN structure, which explains the origins of the broad spectrum of the undulated structure at approximately 450 nm in SEM-CL [23].

High-resolution STEM (HR-STEM) images with chemical information were obtained to examine the difference in emission wavelengths from the flat and slanted MQWs. Atomic resolution images of the MQW region are shown in Figure 3b–d, and the crystallographic orientation was determined from the HR-STEM and Fourier-transformed pattern, which is shown in Figure 3g.

The periods and the thickness of the slanted MQWs were similar to those of the flat MQWs with less than a 0.1 nm difference, and MQW thicknesses were considered to be the same within the margin of error. By comparing the atomic arrangement, HR-STEM image, and Fourier transformed pattern in Figure 3e–g, respectively, the slanted MQW plane was confirmed to be $\left(01\bar{1}1\right)$. Figure 3h shows the STEM-EDS analysis results from the undulated region. Quantitative EDS analysis revealed that the indium concentration from the sidewall $\left(01\bar{1}1\right)$ facet MQWs was approximately 4% higher than that in the flat $\left(11\bar{2}2\right)$ MQWs. This difference in indium concentration was also confirmed by the higher z-contrast in the slanted MQWs as can be seen in Figure 3i.

## 4. Discussion

According to Vegard’s law, the *InGaN* ternary alloy’s bandgap depends on the indium composition, *x*, and a bowing parameter, *b*.
(1)$$E_{InGaN}\left(x\right)=\left(1-x\right)E_{GaN}+xE_{InN}-bx\left(1-x\right)$$

Because the thicknesses of the QWs were almost the same, it was assumed that the difference in bandgap energy *Eg* originated solely from the indium concentration. The calculated average *Eg* of the slanted $\left(01\bar{1}1\right)$ MQWs from the emission wavelength was 2.68 eV, yielding a composition ratio of In_0.16_Ga_0.84_N in the MQW, while the flat $\left(11\bar{2}2\right)$ MQWs exhibited an *Eg* of 2.84 eV with indium incorporation of 0.12%, i.e., In_0.12_Ga_0.88_N. The difference in indium concentration between the flat and the slanted MQWs was in good agreement with the calculations based on the observed emission wavelengths. The values used in the estimation were E*_GaN_* = 3.4 eV, E*_InN_* = 0.77 eV, and *b* = 1.43 eV [30]. Based on the observations and calculations, it is believed that the $\left(01\bar{1}1\right)$ plane accommodated higher indium incorporation, resulting in a longer wavelength emission. According to a previous study [31], the relative incorporation probability of indium in different planes was:$$\left(10\bar{1}1\right)>\left(11\bar{2}2\right)>\left(0001\right)\;≅\left(20\bar{2}1\right)\;≅\left(10\bar{1}2\right)$$

This is consistent with a report on the semi-polar facet properties of a GaN hexagonal annular structure [32].

Some conflicting results have also been reported in terms of the characteristics of planar-grown semi-polar GaN [33] and indium incorporation probability increasing from the tilted crystallographic plane from the *c*-plane towards the *a*- and *m*-planes [34]. Based on the observations presented herein, the selection of the proper GaN growth plane can increase indium incorporation if the growth conditions are fixed or exhibit limited variations.

## 5. Conclusions

The undulated structures originating from the growth of GaN/InGaN MQWs on *m*-plane Al_2_O_3_ were analyzed using TEM-CL, HR-STEM, and EDS. The growth planes of MQW followed the contour profile of the GaN undulation, leaving flat $\left(11\bar{2}2\right)$ facets and slanted $\left(01\bar{1}1\right)$ facets with indium incorporations of 12% and 16%, respectively. Even though the growth planes developed facets of $\left(11\bar{2}2\right)$ and $\left(01\bar{1}1\right)$, the same MQW thickness was observed regardless of the growth plane. The luminescence peak from flat MQWs was observed at approximately 416 nm, whereas that of the slanted MQWs showed two maxima near 458 and 478 nm, whose ratio can be used to explain the non-uniform luminescence and anisotropic emission of LEDs. The combination of the nano-scale luminescence and the high-resolution TEM identifies the variation of the luminescence wavelengths from different growth planes and indium concentrations, which results in the broadening of the visible spectrum convoluted with multiple wavelengths and cannot be identified in the macro-scale electro-luminescence. The results presented herein are expected to assist in the analysis of the characteristics of the semi-polar GaN epitaxial layers and the development of high-efficiency GaN LEDs.

## Figures and Tables

**Figure 1 nanomaterials-13-01946-f001:**
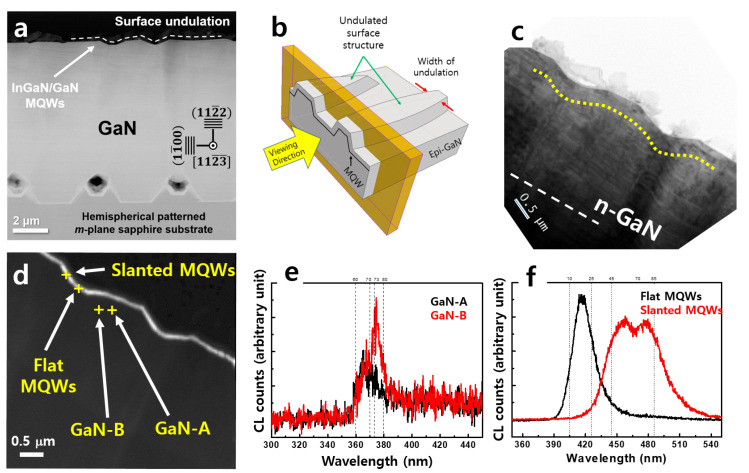
(**a**) Cross-sectional ADF STEM image of the $\left(11\bar{2}2\right)$ semipolar InGaN/GaN LED structure grown on the HPSS. (**b**) Schematic of the surface-undulated sample with the viewing direction for TEM analysis. (**c**) Cross-sectional BF image of the analyzed region, where the location of the MQW is marked with yellow dotted line. (**d**) a panchromatic CL map (λ: 350–550 nm) of the (**c**). (**e**) CL point spectra of GaN-A and GaN-B labeled with cross-hairs in (**d**). (**f**) CL point spectra of the two cross-hair points of flat and slanted MQWs, which is labeled in (**d**).

**Figure 2 nanomaterials-13-01946-f002:**
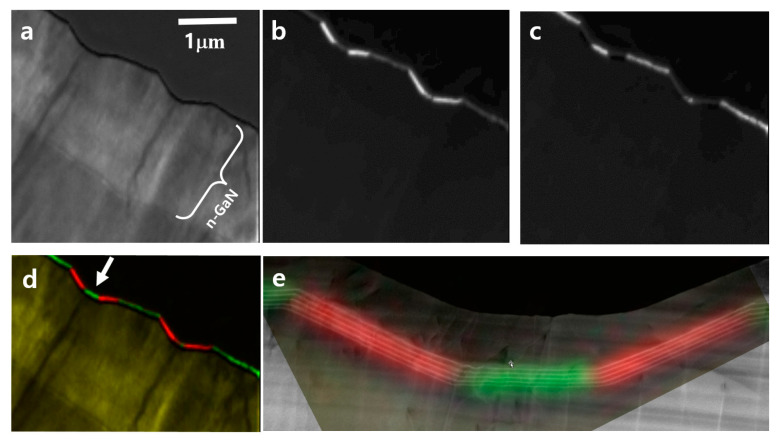
Monochromatic CL intensity mapping from the emission of (**a**) NBE and BSFs (λ: 360–370 nm), (**b**) flat MQWs (λ: 410–425 nm), (**c**) sidewall MQWs (λ: 445–485 nm). (**d**) Red-green-yellow (RGY) composite color CL map. Red indicates the emission of sidewall MQWs, green color represents the emission of flat MQWs, and yellow color shows the emission of the NBE and BSFs. (**e**) An enlarged image of the MQW region marked with an arrow in (**d**), superimposed on its STEM image.

**Figure 3 nanomaterials-13-01946-f003:**
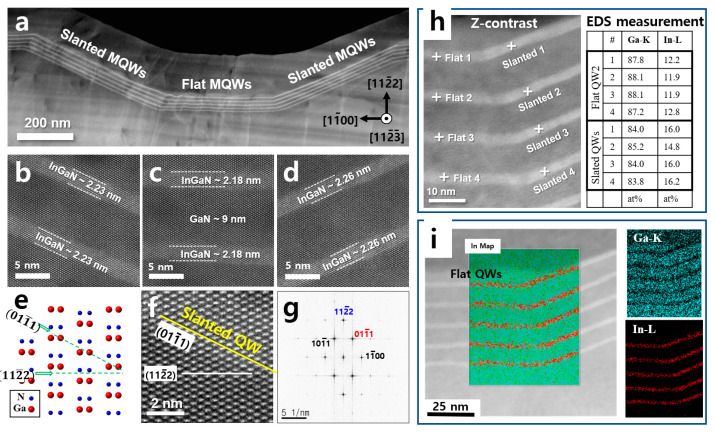
(**a**) The ADF-STEM image of the MQWs of the region indicated with an arrow in Figure 2d. (**b**–**d**) High-resolution ADF-STEM images of the left sidewall, Flat region, and right sidewall MQWs are shown in (**a**), respectively. (**e**) Schematic diagram of an atomic arrangement in semipolar $\left(11\bar{2}2\right)$ GaN. (**f**) An enlarged image of (**b**). (**g**) fast Fourier transformed (FFT) image of (**f**). (**h**) STEM image with a table showing STEM-EDS measurement results. Relative amounts of Ga and In elements are compared from the flat and the slanted region MQWs. (**i**) STEM-EDS mapping results of the flat and the slanted region MQWs.

## Data Availability

Not applicable.

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
