# Peer review of "Correlation between the Surface Undulation and Luminescence Characteristics in Semi-Polar 112¯2 InGaN/GaN Multi-Quantum Wells"

_nanomaterials, 2023, doi:10.3390/nano13131946_

Round 1

Reviewer 1 Report

Given the present interest in GaN-based research, this work is a timely contribution to this interesting field. However, the Authors must address the following issues before I can recommend the publication of this paper in Nanomaterials.

1. It is known that the composition x of InxGa1-xN is important for its physical properties. For example, see Lin et al., J. Phys. Appl. 97, 046101 (2005). The Authors should say something regarding low x and its impact on their work, please. 

2. I understand that in this field, people often use (11-22) and things like that. Couldn't they use (1 1 2bar 2) instead? Please advise. In figure 1, they did use this, right?

3. Are there scale bars in Fig 2 (b) - (e), please?

4. In Eq. (1), InGaN should not be italicized, please. Please kindly check their manuscript.

5. Finally, I am not sure whether Eq. (2) makes sense?

Author Response

Attached the pdf file of the response.

Reviewer 2 Report

This work systematically studied the crystallographic and optical properties of the surface undulation of semi-polar (11-22) InGaN/GaN multi-quantum wells grown on HPSS substrate by MOCVD. The authors identified different emission spectra at different regions of the undulating structures. The emission wavelengths, corresponding In composition, and the possible origins were discussed. The paper is informative and has good presentation, but please address the following comments before it can be recommended for publication. Thank you. 

1. (line 67-68) Surface undulation could be very sensitive to growth conditions. I suggest including more growth parameters that the authors deem necessary for others to replicate the results. For example, source types, substrate temperature, V-III ratio, etc. 

2. (line 68-70) Please specify the doping levels of the n-type and p-type GaN layers, and the detailed structure of the InGaN/GaN multi-quantum wells, i.e. thickness and composition. 

3. (Figure 1) Please include a planar view image of the sample surface. It can be an SEM image or an AFM image. It will provide the readers with critical information about the undulating features, such as their dimensions, density, and spacial distribution. 

4. (Figure 1d) Please improve the presentation of this figure. The arrows are not pointing at the correct crosshair positions. 

5. (Section 5. Conclusions) Please add more discussions about the implications and potential impact of this study in the conclusion section. How would the difference in emission and In incorporation affect the application of semi-polar GaN for optoelectronics? To what extent would it subside the benefit of using (11-22) plane GaN? 

Author Response

Attached the response file.
